# Essential Oils and COVID-19

**DOI:** 10.3390/molecules27227893

**Published:** 2022-11-15

**Authors:** Mahmoud Fahmi Elsebai, Marzough Aziz Albalawi

**Affiliations:** 1Department of Pharmacognosy, Faculty of Pharmacy, Mansoura University, Mansoura 35516, Egypt; 2Department of Chemistry, Alwajh College, University of Tabuk, Tabuk 47512, Saudi Arabia

**Keywords:** essential oils, COVID-19, SARS-CoV-2, eucalyptol, garlic oil, natural products

## Abstract

Herbal products are a major source of herbal medicines and other medicines. Essential oils have shown various pharmacological activities, such as antiviral activity, and therefore are proposed to have potential activity against SARS-CoV-2. Due to their lipophilicity, essential oils can easily penetrate the viral membrane and cause the viral membrane to rupture. In addition, crude essential oils usually have many active constituents that can act on different parts of the virus including its cell entry, translation, transcription, and assembly. They have further beneficial pharmacological effects on the host’s respiratory system, including anti-inflammatory, immune regulation, bronchiectasis, and mucolytics. This review reported potential essential oils which could be promising drugs for COVID-19 eradication. Essential oils have many advantages because they are promising volatile antiviral molecules, making them potential drug targets for the prevention and treatment of COVID-19, whether used alone or in combination with other chemotherapeutic drugs. The aim of the current review is to shed light on the potential essential oils against enveloped viruses and their proposed activity against SARS-CoV-2 which is also an enveloped virus. The objectives were to present all data reflecting the promising activities of diverse essential oils against enveloped viruses and how they could contribute to the eradication of COVID disease, especially in indoor places. The data collected for the current review were obtained through the SciFinder database, Google scholar, PubMed, and Mendeley database. The data of the current review focused on the most common essential oils which are available in the pharmaceutical market and showed noticeable activities against enveloped viruses such as HSV and influenza.

## 1. Introduction

The COVID-19 pandemic is grim globally. The manifestations of COVID-19 infection vary, ranging from asymptomatic disease to severe acute respiratory infection. Fever, dyspnea, dry cough, loss of appetite, fatigue, myalgia, and taste and smell dysfunction are the main systemic symptoms. Suppressed regulatory T cells (CD8+ and CD4+), Natural Killer (NK) cells, macrophages/monocytes, and increased proinflammatory cytokines such as IL-1, IL-2, IL-6, TNF-α is to weaken the immune system. Respiratory failure, septic shock, and/or multiple organ failure were recognized in critically ill patients [1,2,3,4,5].

Different ways of treatment against COVID-19 were established such as antiviral treatments (including remdesivir, molnupiravir), and monoclonal antibodies, in addition to symptomatic treatment such as using antipyretic, anticoagulant, and corticosteroids. Due to the epidemic nature of the coronavirus, alternative and complementary methods should be considered [6,7].

Essential oils have been broadly used in traditional and folk medicine due to their diverse biological activities including anti-inflammatory, antioxidant, immunomodulatory, antimicrobial (antiviral, antifungal, antibacterial), antirheumatic, expectorant, antitussive, and sedative properties [8,9]. Essential oils (EOs) are volatile secondary plant metabolites from diverse chemical classes including hydrocarbons (=terpenes, that occur practically in all EOs and they can be classified biogenetically into monoterpenes, sesquiterpenes, diterpenes, …etc), oxygenated hydrocarbons (=terpenoids, such as alcohols, phenols, aldehydes, ketones, esters, phenyl methyl ethers, acids, oxides, peroxides), non-terpenic compounds biosynthesized by the phenylpropanoids pathway (such as cinnamaldehyde, eugenol, safrole), cyanate, isothiocuanate, sulfur, and nitrogenous compounds [8,10]. EOs are mainly produced by more than 17,500 species of certain plants belonging to the angiosperm families, such as Alliaceae, Asteraceae, Umbelliferae, Cruciferae, Myrtaceae, Lamiaceae, Zingiberaceae, and Rutaceae [11,12]. They are biosynthesized from the fruits, flowers, leaves, and roots of different aromatic plants and stored in specialized tissues such as glandular hairs, glands, oil cells, oil ducts, and oil receptacles [10].

The clinical applications of EOs are mainly directed to midwifery, cancer and palliative care, elder care, and mental health such as in the case of depression. The pharma market is full of many EOs pharmaceutical products used for the treatment of insomnia, cough, asthma, urinary tract infections, and kidney stones among others. Aromatic essential oils are the basis of aromatherapy.

These EOs have shown potential activities against a broad spectrum of viruses, such as coronavirus, influenza virus, human immunodeficiency virus (HIV), human herpesviruses (HSV1 and HSV2), avian influenza, and yellow fever virus [10,13,14,15,16,17,18]. This current review discusses the published data on the possible contribution of EOs in the treatment and prevention of COVID-19 as EOs have characteristic advantages, namely their characteristic volatility and broad antiviral activities; this could add therapeutic value to the drug candidate. 

## 2. Essential Oils with Potential Activities against SARS-CoV-2

### 2.1. Eucalyptus Oil

Eucalyptus oil and its main active component, eucalyptol (=1,8-cineole) have been reported to exhibit anti-inflammatory, and antiviral activities, among others [19,20]. Eucalyptol is isolated from *Eucalyptus* species and found in camphor tree, tea tree, rosemary, common sage, and *Cannabis sativa* [21]. Eucalyptol is a cyclic monoterpenoid oxide or ether (Figure 1), colorless oily with a camphoraceous odor and pungent taste, occurs in many essential oils, and represents a major product in eucalyptus oil (more than 70%) [21,22].

Eucalyptus oil demonstrated antiviral activities against different viruses including herpes simplex viruses HSV1 and HSV2, enveloped mumps viruses (MV), influenza A (H1N1) virus, rotavirus Wa, coxsackie B, poliovirus, and echovirus 6. Additionally, eucalyptus oil exhibited disinfection properties and inhibited the proliferation of viruses on different filters and utensils [13,23,24]. In an ex vivo model of rhinosinusitis and in human stem cells, eucalyptol exhibited an antiviral response stimulation both by decreasing the activity of the proinflammatory NF-κB and by enhancing the activity of antiviral transcription factor IRF3 [25], and in mice, it has shown a protective effect against influenza-virus-induced pneumonia [26].

Based on the antiviral activity of eucalyptol and eucalyptus oil toward respiratory viruses, many researchers have used molecular docking techniques and in vitro assays to investigate the antiviral bioactivity of eucalyptus oil and its major components “eucalyptol and jensenone” (Figure 1) against SARS-CoV-2 [27,28]. The reported docking studies of both eucalyptol and jensenone have shown the antiviral potential to inhibit the Mpro of SARS-CoV-2 [27]. 

### 2.2. Garlic Oil

*Allium sativum* (garlic) is a functional food well-known for its anti-inflammatory, antioxidant, immunomodulatory, antimicrobial, antitumor, neuroprotective, digestive system protective, hepatoprotective, anti-obesity, anti-diabetic, cardiovascular protective, and antimutagenic properties. Garlic contains many bioactive components such as organo-sulfur compounds, polysaccharides, phenolic compounds, and saponins. The organo-sulfur compounds (OSCs), such as allicin, alliin, S-allyl-cysteine, ajoene, diallyl sulfide, diallyl disulfide, and diallyl trisulfide (Figure 2) are major bioactive constituents in garlic [18,29]. 

The literature data show that garlic and its active OSCs have a broad spectrum of antiviral activity against several viruses that cause widespread infections including coronavirus, SARS-CoV, influenza, parainfluenza, porcine reproductive and respiratory syndrome virus, measles, rhinovirus, rotavirus, HSV, enterovirus, coxsackie B, HIV, porcine rotavirus, hepatitis A virus, dengue virus, cytomegalovirus, and vaccinia virus [18].

Allicin is one of the main OSCs of allium and the principal components responsible for the pharmacological properties of garlic such as the antiviral [30], anti-inflammatory [31], antioxidant [32], and immunomodulatory activities [33,34]. Pre-clinical studies showed that sulfur derivatives of allicin such as allitridin, ajoene, garlicin, and DAS, also have potential antiviral [35,36], and immune-enhancing potential [37,38]. Clinical trials of various over-the-counter garlic formulations have shown protective effects against some human viral infections, including viral hepatitis, influenza, and warts. Enhancing the immune system was claimed to be responsible for these effects [18]. Garlic appears to counteract immune system dysfunction in COVID-19-infected patients. Therefore, garlic may protect against COVID-19 infection by boosting immune system cells and reducing the production of proinflammatory cytokines.

### 2.3. Other Aromatic Herbs

The EOs of *Laurus nobilis* are used in folk medicine for the treatment of rheumatoid diseases. The in vitro antiviral activities showed that *L. nobilis* EO exerted an interesting inhibitory activity against SARS-CoV replication (IC_50_ = 120 µg/mL). The GC-MS analysis demonstrated that its main volatile constituents are α-pinene, β-pinene, β-ocimene, (Figure 3), and eucalyptol. EOs of *Juniperus oxycedrus* ssp. oxycedrus, of which α-pinene and β-myrcene are the main components, exhibited antiviral activity against HSV1 (IC_50_ value 200 µg/mL) [39]. 

qPCR and immunoblot analysis showed that lemon and geranium oils have potent ACE2 inhibition in epithelial cells. GC-MS analysis showed that geraniol, neryl acetate, and citronellol (Figure 4) were the main components of geranium oil, and limonene (Figure 4) was the major bioactive constituent of lemon oil. Treatment with limonene and citronellol effectively downregulated ACE2 expression in epithelial cells suggesting that lemon and geranium EOs and their derivatives are promising natural antiviral agents against SARS-CoV-2/COVID-19 [40]. 

EOs from different medicinal aromatic plants such as *Mentha* spp., *Citrus* spp., *Illicium* spp., *Hyssopus officinalis*, tea trees, mayweeds, *Pinus* spp., chamomile, *Santalum* spp., ginger, thymes, and other aromatic plants with antiviral activity were well reported by some scientific research [13,41,42].

## 3. Mode of Action of Essential Oils

Enveloped viruses are sensitive to EOs [10]. The major mode of action for EOs is through the prevention of viral attachment and entry, which is attributed to their lipophilic characters, enabling them to interact with the lipid bilayer within the viral envelope. As a result, the fluidity and permeability of the cytoplasmic membrane were altered, and at an increased concentration, the membranes were ruptured [17,41]. Using a plaque reduction assay on RC-37 cells, the EOs from thyme, ginger, chamomile, anise, sandalwood, and hyssop exhibited an inhibitory effect against HSV2 mainly before adsorption by interacting with the viral envelope [42]. The concentrations at which EOs exert their antiviral activity are typically much lower than cytotoxic concentrations, suggesting that the virion envelope is more sensitive to EOs than host cell membranes [10]. *Eugenia caryophyllus* and eugenol directly inactivated the particles of HSV standard strains [43]. EOs from tea tree, thyme, and eucalyptus and their main isolated components α-pinene, α-terpinene, α-terpineol, terpinen-4-ol, ɣ-terpinene, citral, *p*-cymene, eucalyptol, and thymol (Figure 5) reduced HSV1 viral infectivity by >96% (for the total EO), and >80% (for individual monoterpene). Both EOs and their monoterpenes showed higher anti-HSV1 activity by directly inactivating free virus particles than when these drugs were added to host cells before infection or after cell entry [13]. However, carvacrol acts directly on the viral capsid of murine norovirus and subsequently on nucleic acids [44]. 

OSCs exhibited diverse modes of antiviral activities including: inhibition of the viral cell cycle (such as blocking viral entry into host cells, inhibiting viral reverse transcriptase, viral replication, and viral RNA polymerase), improving the host immune reaction, and reducing cellular oxidative stress [18]. OSCs are highly reactive to the thiol groups (SH) which are available in various active viral enzymes and proteins important for the viral life cycle [45]. 

During the COVID-19 infection, there is a decrease in the number of the T lymphocytes helper (CD4+), and the cytotoxic T cells (CD8+) which are two major types of T cells and are crucial soldiers of the immune system, and suppression of the NK cells which also possess cytolytic activity against virus-infected and tumor cells [46,47], resulting in a decrease of IFN-γ production [4,5]. Immunomodulatory activity enhances the innate immune system through NK and macrophage cells and also enhances adaptive immunity through B cells, T cells, and anti-inflammatory cytokines leading to viral clearance [18,48]. EOs appear to be able to modulate erratic immunological responses. Garlic compounds potentially decreased the expression of proinflammatory cytokines and reversed the immune abnormalities to normal levels. The absolute numbers of CD4 + T cells, macrophages, and dendritic cells (DC) were significantly elevated in allicin-treated mice with malaria infection [49]. Many in vitro and in vivo studies showed that garlic possessed immunomodulating properties which were attributed to the OSCs, lectins, and water-soluble inulins [36,38,50,51,52,53,54]. Furthermore, many pieces of research revealed that OSCs exhibited anti-inflammatory activities involved in viral infections [5,30,37,55,56,57,58]. 

Oxidative stress induced by viral infection plays a role in the viral life cycle and pathogenesis of the viral disease (by impairing the host immune system) to the activation of host antioxidant pathways including Nrf2 [37,59] which significantly decreased the intensity of cytokine storms in COVID-19 patients [24]. In lung MRC-5 cells, diallyl sulfide (DAS) induces Nrf2 activation by triggering p38/ERK-signaling pathways [60,61]. In diet-induced obese mice, alliin decreased the expression of proinflammatory cytokines [33,62] and reduced inflammatory markers [62]. 

Eucalyptol has different pharmacological potentials against various respiratory ailments including sinusitis, pharyngitis, bronchitis, and a nasal decongestant effect (as in Vicks VapoRub™) [63,64]. Eucalyptol and eucalyptus oil were safe and effective in many clinical trials and its inhalation, by blocking cytokines release, exerted an anti-inflammatory influence; hence, it can be effectively used in chronic obstructive pulmonary disease (COPD), fever, flu, bronchitis, sinusitis, respiratory infections, and asthmatic patients [20,63,65,66,67,68,69,70]. Eucalyptol inhibited the release of proinflammatory cytokines TNF-α and IL-1β from lipopolysaccharide (LPS)-stimulated monocytes [63], and significantly inhibited the NF-κB p65 gene promoter in LPS-stimulated human cell lines thereby reducing inflammation [71]. In vivo studies by Shao et al. in healthy rats, have shown that low doses of eucalyptol (≤100 mg/kg) stimulated immune responses, whereas high doses of eucalyptol impair respiratory immune function and immunity, possibly through B cell modulation, which increases immune complexes, inflammatory intermediates, and plasma cells [22].

Using an LPS-induced acute lung injury in the mouse model, cinnamaldehyde markedly reduced lung wet/dry ratio and pulmonary edema. It inhibited also significantly macrophages, total cell number, neutrophils, and, in the bronchoalveolar lavage fluid, there were reduced levels of inflammatory cytokines such as IL-1β, IL-6, IL-13, and TNF-α [72]. 

In a mouse model of elastase-induced pulmonary emphysema, carvacrol (Figure 6) reduced alveolar enlargement, macrophage recruitment, and the levels of IL-6, IL-8, IL-17, and IL-1β [73]. Carvacrol has antiviral activities against HSV1, acyclovir-resistant HSV1, human rotavirus (RV), and human respiratory syncytial virus (HRSV) [74]. 

Menthol-rich plant extracts (such as *Mentha piperita*, Labiatae) (Figure 6) have been used in folk medicine to treat respiratory ailments such as nasal congestion, rhinitis, dyspnea, and COPD [75,76]. In rat models, menthol demonstrated immunomodulatory, anti-inflammatory, and gastroprotective properties. Treatment with menthol significantly reduced the levels of proinflammatory cytokines IL-1, IL-23, and TNF-α in rats [77,78].

Eugenol (Figure 6) exhibited antiviral activities against HSV1 and 2 [79], anti-inflammatory properties, and protected the lungs toward LPS-induced acute injury where it downregulated the expression of proinflammatory cytokines TNF-α and IL-6 and inhibited the recruitment of leukocytes into the lung [80].

Inhalation of vapors containing essential oils with antiviral, anxiolytic, decongestant, and other properties, may further help to promote mucociliary clearance and inactivate or inhibit virions where they initially reside and further support innate and acquired immune defense [81,82,83].

The selective index for different essential oils and their components was reported [13].

## 4. Activities of EOs Compared to That of Their Principal Components

It is generally accepted that the bioactivities of EOs are determined by their chemistry [17]. Eugenol, a principal constituent of *Cinnamomum zeylanicum* EO, is a potent anti-H1N1 [84], suggesting that the antiviral efficacy of the EO could be attributed to its major constituents. Both *Eucalyptus globulus* and *Salvia officinalis* contain the major constituent eucalyptol, but the former showed strong anti-H1N1 activity (IC_50_ < 3.1 μg/mL) whereas the latter did not [84], indicating that the minor components of EOS may be more bioactive than the major ones. Carvacrol was less efficient than its original Mexican oregano EO against HSV1, respiratory syncytial virus, and bovine viral diarrhea virus, while it efficiently inhibited rotavirus on which the Mexican oregano EO showed no inhibitory effect at all at the concentrations tested (25–3200 µg/mL) [85].

## 5. Advantages of Essential Oils as Antiviral Drugs

Lungs and air pathways are the target organ for the start of SARS-CoV-2 infection and thus, it is advantageous that the EOs are mainly administered by inhalation and produce their direct effect in the airways and lungs. This gives a good chance for the EOs to distract the binding between SARS-CoV-2 spike proteins and their cognate ACE2 receptors on the lung’s parenchyma.

EOs with prospected anti-COVID-19 activity can be delivered and concentrated into the lung, and significantly reach the site of action to achieve their tasks [86].

Vapor generation of EO has a better antibacterial effect than liquid EO. In the aqueous phase, lipophilic molecules aggregate to form micelles, which inhibit the attachment of EO to microorganisms while the vapor phase of EO acts, for example, specifically on fungi due to their surface growth [87]. 

Inhalation is the preferred route of administration for many drugs that act directly on the respiratory tract, especially in conditions such as COPD and asthma [88]. 

A major advantage of inhaled drugs is the lack of first-pass metabolism, which increases bioavailability [86].

Volatile molecules, such as EO, are characterized by high vapor pressures and low molecular weights, so they are readily cleared from the lungs and exhaled, providing local, even non-specific, anti-inflammatory and virucidal effects. enhances its pharmacological effects [86,89]. This property enables eucalyptol to be concentrated in the lungs, especially in the lower respiratory tract and thus exerting its loco potential virucidal effect [16]. 

Due to certain chemical–physical properties of some Eos such as eucalyptol, it is adequately concentrated in the lungs through lung exhalation and thus performing its pharmacological effects in lower doses, e.g., in vitro antiviral activity of aerosolized cineole was performed in two viral model systems of the aerosolized influenza A strain NWS/G70C (H11N9), simulating hazardous bioaerosols in outdoor and indoor environments [23]. After 15 s of aerosolization and 5 min of exposure, 99% viral inactivation was achieved. The in vitro antiviral effect of eucalyptol was measured and eucalyptol exhibited an antiviral IC_50_ equal to about one-fourth of the maximum nonlethal dose obtained in the herpes simplex virus cytotoxicity test [10] and about a sixth of the maximum nonlethal dose for infectious bronchitis virus (IBV) [90]. 

Asymptomatic individuals are known to play a critical role in the spread of SARS-CoV-2 [91]. Eucalyptol is virucidal both when administered orally to the lower respiratory tract to avoid the onset of COVID-19 etiology and when inhaled to the upper respiratory tract to avoid spreading the virus [92].

Whole EOs have multiple advantages over purified components, such as a low probability of selecting for synergistic effects of various ingredients, multiple pharmacological activities, and antimicrobial resistance [93]. Additionally, the combination of EOs might be a good strategy for fighting coronaviruses as they are more effective against foodborne pathogens and spoilage bacteria [94,95]. EO ingredients may act synergistically to enhance the effects of other antiviral drugs, or they may relieve COVID-19 symptoms [96], e.g., 1., combinations of eucalyptol and oseltamivir increased the number of survivors, improved lung parameters (viral titers, lung index, and pathology) and reduced cytokines (IL-1β, IL-10, TNF-α) expression in the lung [97], e.g., 2., the combination of *Melissa officinalis* EO and oseltamivir enhanced the effect of oseltamivir against avian influenza A virus (H9N2) [98], e.g., 3., germacrone plus oseltamivir exhibited additive anti-influenza virus effects in both in vivo and in vitro [99], and, e.g., 4., piperitenone oxide (the main component of *Mentha suaveolens* EO) plus acyclovir also exhibited synergistic activity against HSV1 [15]. 

Essential oils are present everywhere in our life, such as perfumes, air refreshers, aromatic water, and medicinal drugs, and many of them are used in aromatherapy. Therefore, based on this and on their antimicrobial activities, it is strongly advisable to direct them to the disinfection of closed places in which respiratory viruses such as coronaviruses proliferate.

## 6. Conclusions

Natural products are promising lead materials for drug discovery. Herbs and their essential oils have been used for different purposes since the beginning of human history. Their beneficial properties have been used in attracting the attention of other people, fragrances, masking unpleasant odors, aromatherapy, cosmetics, and more. Essential oils are natural products mainly derived from a variety of plants, represent a rich source of bioactive drugs, and have become the basis for the development of new drugs for the enrichment of the pharmaceutical industry, such as the antiviral drug oseltamivir (Tamiflu^®^) that originated from anethole EO of star anise. Many of the commercially available EOs have potential antimicrobial activities and they appear to be promising alternatives to synthetic compounds, particularly due to the emerged microbial resistance which necessitates the development of novel antimicrobial agents [95,100,101,102,103].

Essential oils have many advantages: small molecules, with high lipophilicity, faster onset of action, virucidal properties at lower toxic concentrations, not toxic at therapeutic concentration, high vapor pressure, many of them can be orally administered, pulmonary exhalation and lower respiratory tropism, deeper delivery in lungs, pulmonary delivery, reduced first-pass metabolism, easy formulation and synthesis, cheap preparation, can be indicated as prophylactic agents in the early stages of viral pneumonia and in combination with other therapies. Eucalyptol’s diverse pharmacological properties and its physicochemical properties make it a potential drug candidate for the treatment and prevention of COVID-19. Based on the current knowledge, a combination of EOs (with known pharmacokinetics and pharmacodynamics) and chemical agents may be a more viable and effective approach to combat the COVID-19 virus pandemic, and thus further research is needed in this regard. 

The volatility of EOs facilitates their virucidal activity by facilitating their diffusion into the lungs which are the main targets attacked by SARS-CoV-2. The lungs and the air pathways are the target organ of SARS-CoV-2 infection 

In summary, EOs are promising inhaled volatile agents for chasing COVID-19 by both direct virucidal effects and other pharmacological properties such as anti-inflammatory, immunomodulatory, and antioxidant properties. Therefore, animal models with subsequent clinical studies should be considered.

## Figures and Tables

**Figure 1 molecules-27-07893-f001:**
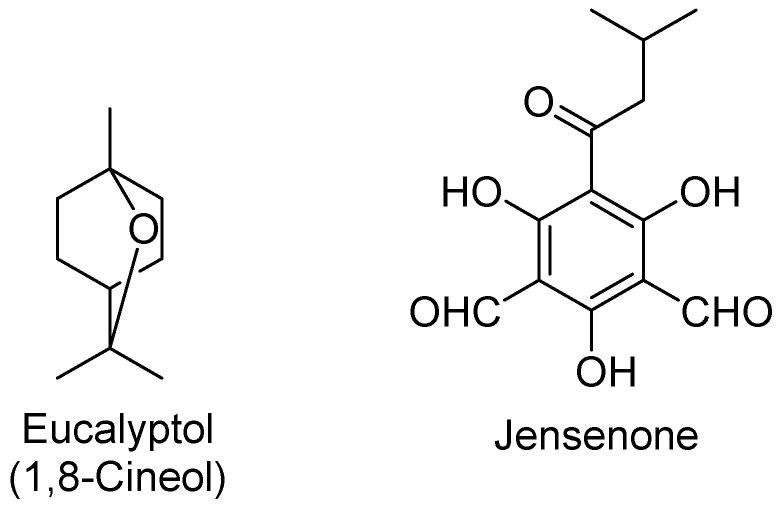
Structures of eucalyptol and jensenone, the active components of eucalyptus oil.

**Figure 2 molecules-27-07893-f002:**
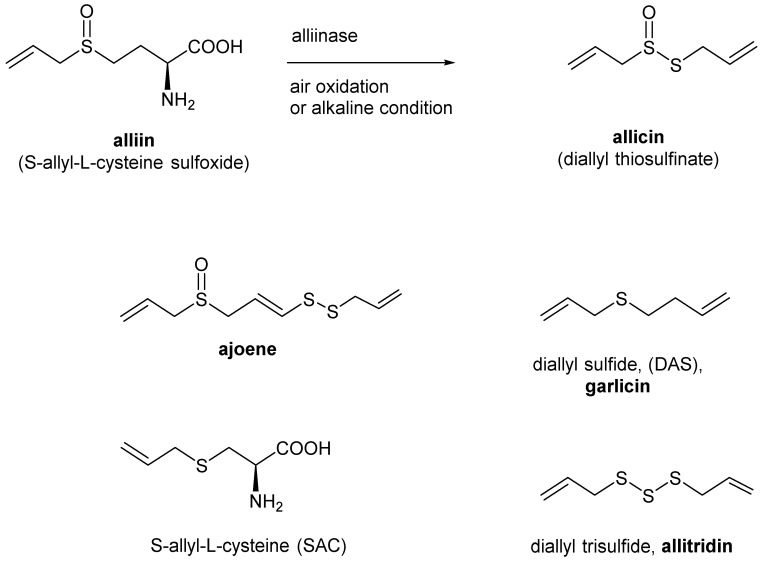
Structures of the organo-sulfur compounds, the active components of garlic oil.

**Figure 3 molecules-27-07893-f003:**
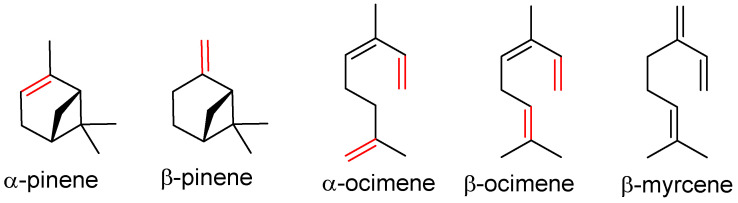
Structures of the active components from *Laurus nobilis* oil and *Juniperus oxycedrus* ssp. Oxycedrus oil.

**Figure 4 molecules-27-07893-f004:**
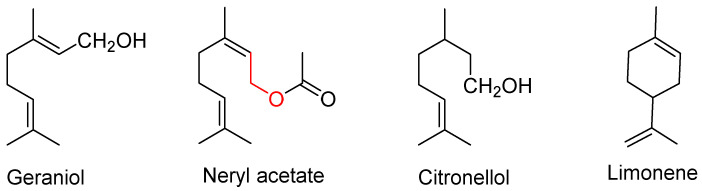
Structures of the active components from lemon and geranium oils.

**Figure 5 molecules-27-07893-f005:**
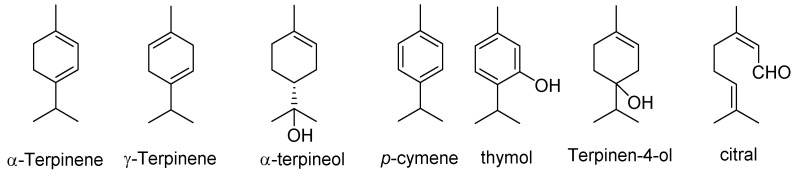
Structures of further essential oils of tea tree, thyme, and eucalyptus oils.

**Figure 6 molecules-27-07893-f006:**
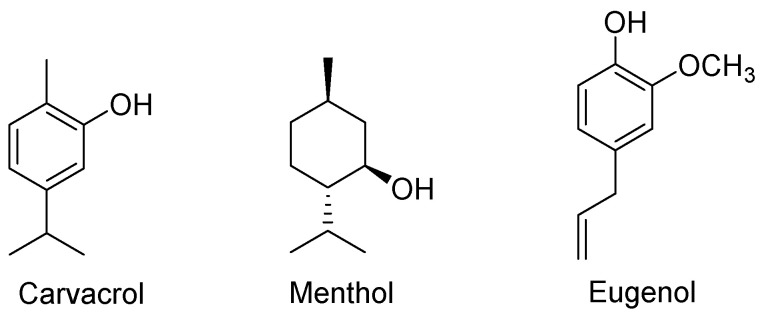
Structures of further essential oils.

## Data Availability

Not applicable.

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
