# Peer review of "Essential Oils and COVID-19"

_molecules, 2022, doi:10.3390/molecules27227893_

Round 1
Reviewer 1 Report
Presended review article summarizes the activities of essential oils against viruses with the major focus on SARS-COV2 virus. The topic is interesting and worth to publish in Molecules, however, I would suggest some improvements.
1. The article contains almost no specific data from the primary sources, it would be great if there is a mentioned activity to add IC50, ED50 or other value in order to be able to compare the essential oins. In addition, when possible - articles with docking etc. - it would be great to mention the molecular targets (proteins etc.) that are inhibited.
2. The article deserves more involvement from the side of the authors - they should express their opinions on the field of this review and add some future perspectives what they believe is the right direction of this kind of research. A review article should help others to orient themselves in the topic and not just to pile a number of article references into one text.
Minor comment: milliliter is mL not ml in english.
In my opinion, this article is worth to publish (and I would be use it and citing it) after changing it into a critical review containing the authors own ideas and opinions.
Author Response
Reviewer 1
Presented review article summarizes the activities of essential oils against viruses with the major focus on SARS-COV2 virus. The topic is interesting and worth to publish in Molecules, however, I would suggest some improvements.
- The article contains almost no specific data from the primary sources, it would be great if there is a mentioned activity to add IC50, ED50 or other value in order to be able to compare the essential oils. In addition, when possible - articles with docking etc. - it would be great to mention the molecular targets (proteins etc.) that are inhibited.
Thanks for this notification, we have not added such like numerical values since they are already published in some articles, for example, reference Phytotherapy Research 2010, 24, 673–679, doi:10.1002/ptr.2955, titled: “Comparative Study on the Antiviral Activity of Selected Monoterpenes Derived from Essential Oils”
Anyway, at page 11 line 193, I did add the sentence: “The selective index for different essential oils and their components were reported [13]”
Regarding the molecular targets using the molecular docking, we added in page 5 line 59 “The reported docking studies of both eucalyptol and jensenone have shown the antiviral potential to inhibit Mpro of SARS-CoV-2 “
- The article deserves more involvement from the side of the authors - they should express their opinions on the field of this review and add some future perspectives what they believe is the right direction of this kind of research. A review article should help others to orient themselves in the topic and not just to pile a number of article references into one text.
The following paragraph is added before the conclusion section page 13 line 250: “The essential oils are present in everywhere in our life, such as perfumes, air refreshers, aromatic water, medicinal drugs, and many of them are used in aromatherapy. Therefore, based on these and on their antimicrobial activities, it is strongly advisable to direct them to the disinfection of closed places in which the respiratory viruses such as CORONA viruses proliferate.”
Minor comment: milliliter is mL not ml in English.
Thanks, I did change all ml into mL
In my opinion, this article is worth to publish (and I would be use it and citing it) after changing it into a critical review containing the authors own ideas and opinions.
Thanks a lot, we added the author opinions at the end of conclusion section: “In summary, EOs are promising inhaled volatile agents for chasing COVID-19 by both direct virucidal effect and other pharmacological properties such as anti-inflammatory, immunomodulatory, and antioxidant properties. Therefore, animal models with subsequent clinical studies should be considered.”
Reviewer 2 Report
This article presents relation and role of essential oils in treating Covid 19 which is helping for the future pharmaceutical industry. Before recommending this article for publication, there are some shortcomings for that should be resolve.
Abstract
In the first sentence “Herbal medicine is a major producer of essential oils and other medicines” should be change to Herbal products/ or essential oils are major source of herbal medicines and other medicines.
English of the following sentence must be correct and clear “oils contain a variety of active ingredients that can act synergistically in multiple stages of the virus life cycle”
Line 17 use Covid-19 instead of CORONA.
The abstract is very general, neither methods of data collection or literature review and nor main findings are discussed.
Aim and objective must be discussed in the abstract.
Capitalize first alphabet of the key words
Introduction
Line 24. Replace global war with pandemic or some other suitable word.
Line 31 Authors already said essential oils are = volatile oils, then line 34 is repetition.
Line 37 write complete classification.
In first paragraph write different ways of treatments against Covid 19, also why herbals are better as compared to other by citing relevant literature. The following articles could be helpful.
doi: 10.1002/ptr.6787, doi.org/10.36721/PJPS.2021.34.4.REG.1469-1484.1, https://doi.org/10.1016/j.jsps.2020.06.022,
In third chapter specific uses and applications of the Eos must be discussed in detail.
What kind of role and contributions have been played by EOs in the fight against Covid 19.
Line 87 to 90 add more relevant references and remove repetition of and as well.
As a while the review article is well presented but mostly citations are missing, and sentences are not clearly written.
In conclusion main findings and future recommendations must be provided.
Author Response
Reviewer 2
This article presents relation and role of essential oils in treating Covid 19 which is helping for the future pharmaceutical industry. Before recommending this article for publication, there are some shortcomings for that should be resolve.
Abstract
In the first sentence “Herbal medicine is a major producer of essential oils and other medicines” should be change to Herbal products/ or essential oils are major source of herbal medicines and other medicines.
Thanks, I did change it into “Herbal products are major source of herbal medicines and other medicines”
English of the following sentence must be correct and clear “oils contain a variety of active ingredients that can act synergistically in multiple stages of the virus life cycle”
Thanks, I did change it into “the crude essential oils usually have many active constituents that can act on different parts of the virus including its cell entry, translation, transcription, and assembly.”
Line 17 use Covid-19 instead of CORONA.
Thanks, I did change it.
The abstract is very general, neither methods of data collection or literature review and nor main findings are discussed.
The following paragraph is added to the end of abstract: “The data collection for the current review were obtained through the SciFinder data base, Google scholar, PubMed, and Mendeley data base. The data of the current review focused on the most common essential oils and which are available in the pharmaceutical market and showed noticeable activities against SARS-COV and other enveloped viruses such as HSV and influenza.”
Aim and objective must be discussed in the abstract.
The following paragraph is added to the abstract “The aim of the current review is to shed light on the potential essential oils against enveloped viruses and their proposed activity against SARS-COV-2 which is also an enveloped virus. The objectives were to present all data reflecting the promising activities of diverse essential oils against enveloped viruses and how they could contribute to the eradication of COVID disease specially in the indoor places.”
Capitalize first alphabet of the key words
Thanks, I did it
Introduction
Line 24. Replace global war with pandemic or some other suitable word.
I did change it into: The COVID-19 pandemic is grim globally.
Line 31 Authors already said essential oils are = volatile oils, then line 34 is repetition.
Thanks, I deleted the first “volatile oils”
Line 37 write complete classification.
Thanks, it became now: “hydrocarbons (= terpenes, that occur practically in all EOs and they can be classified biogenetically into monoterpenes, sesquiterpenes, diterpenes, …etc), oxygenated hydrocarbons (= terpenoids, such as alcohols, phenols, aldehydes, ketones, esters, phenyl methyl ethers, acids, oxides, peroxides), non-terpenic compounds biosynthesized by the phenylpropanoids pathway (such as cinnamaldehyde, eugenol, safrole), cyanate, isothiocyanate, sulfur, and nitrogenous compounds.
In first paragraph write different ways of treatments against Covid 19, also why herbals are better as compared to other by citing relevant literature. The following articles could be helpful.
doi: 10.1002/ptr.6787, doi.org/10.36721/PJPS.2021.34.4.REG.1469-1484.1, https://doi.org/10.1016/j.jsps.2020.06.022,
Different ways of treatment against Covid 19 were established such as antiviral treatments (including remdesivir, molnupiravir), and monoclonal antibodies, in addition to symptomatic treatment such as using antipyretic, anticoagulant, and corticosteroids.
These references were cited: doi: 10.1002/ptr.6787, https://doi.org/10.1016/j.jsps.2020.06.022,
In third chapter specific uses and applications of the Eos must be discussed in detail.
I did add this paragraph: “The clinical applications of Eos are mainly directed to midwifery, cancer and palliative care, elder care, and mental health such as in the case of depression. The pharma market is full of many Eos pharmaceutical products used for treatment of insomina, cough, asthma, urinary tract infections, kidney stones among others. Aromatic essential oils are the basis of aromatherapy.”
Dear Reviewer, to discuss the specicifc uses and applications of essential oils, it will need another review to do so, since their uses and applications are plenty.
What kind of role and contributions have been played by EOs in the fight against Covid 19.
So far, there is no serious role and contribution played by Eos in the fight against Covid-19; this is why we are presenting such like data about essential oils to draw the attention towards the promising importance of their use against CORONA virus specially within the indoor places. At the end of introduction, I said: “The current review discusses the published data on the possible contribution of EOs in the treatment and prevention of COVID-19 as EOs have characteristic advantages, namely their characteristic volatility and broad antiviral activities; this could add therapeutic value to the drug candidate.”
Line 87 to 90 add more relevant references and remove repetition of and as well.
I checked out the whole text for “of” and “as well” and removed some of them.
I did add a more relevant reference and updated one (17 instead of 10)
As a while the review article is well presented but mostly citations are missing, and sentences are not clearly written.
Thanks, we checked out again the whole text and edited some sentences.
In conclusion main findings and future recommendations must be provided.
Thanks, the following is added to the conclusion: “In summary, EOs are promising inhaled volatile agents for chasing COVID-19 by both direct virucidal effect and other pharmacological properties such as anti-inflammatory, immunomodulatory, and antioxidant properties. Therefore, animal models with subsequent clinical studies should be considered.”
